# IVTFuse: An Efficient Vision-Language Guided Infrared-Visible Image Fusion Network with Frequency-Strip and Hybrid Pooling Attention Modules

**GPT-4o**

**Zixuan Liu**
Department of Computer Science
Tulane University
zliu41@tulane.edu

**Siavash Khajavi**
Department of Industrial Engineering and Management
Aalto University
siavash.khajavi@aalto.fi

**Guangkai Jiang**
guangkaijiang@gmail.com

**Xinru Liu**
Department of Industrial Engineering and Management
Aalto University
xinru.liu@aalto.fi

## Abstract

Infrared-visible image fusion (IVF) aims to combine complementary thermal and visible information into a single image that is informative for both human observation and computer vision tasks. However, existing fusion methods often struggle to preserve both the fine details and the semantic context of a scene, especially when relying solely on image-based features. We propose **IVTFuse**, a novel vision-language guided fusion network that addresses these challenges by incorporating textual semantic guidance and frequency-aware attention mechanisms. IVTFuse introduces two lightweight modules: *Frequency Strip Attention* (FSA) and *Hybrid Pooling Attention* (HPA), within each modality-specific encoder to adaptively enhance crucial structures and regions. Meanwhile, a text description of the scene is encoded by a pre-trained BLIP model and injected into the fusion process through cross-attention, providing high-level context to guide feature merging. Our architecture is built on efficient Restormer-based transformers and maintains a compact model size, making it feasible for real-time applications. Extensive experiments on standard infrared-visible fusion benchmarks show that IVTFuse outperforms 10 state-of-the-art methods across three public IVF datasets, producing fused images with improved detail and semantic fidelity. The code is available at https://github.com/ZixuanLiu4869/IVTFuse

## 1 Introduction

Image fusion is a critical technique in computer vision that combines information from multiple source images into a single composite, yielding a more informative result for human or machine perception. Among fusion tasks, infrared-visible image fusion (IVF) is especially important in domains like night vision, surveillance, and autonomous driving. By fusing an infrared (IR) image capturing thermal emissions with a visible-light image capturing fine textures and colors, IVF can

produce an output that reveals salient targets (e.g. warm objects in darkness) while preserving the contextual scene details from the visible spectrum. Achieving effective IR-VIS fusion is challenging, however, because the two modalities have very different characteristics and the fusion must retain both low-level details and high-level contextual information.

In recent years, deep learning approaches have driven significant progress in image fusion. Convolutional neural network (CNN) based fusion models have demonstrated strong ability to extract and merge low-level features (edges, textures, intensities) from different modalities. A variety of CNN architectures have been explored for infrared-visible fusion, from encoder–decoder frameworks to densely connected networks and generative adversarial models [1, 2, 3]. These methods effectively blend pixel-level information from IR and visible images, but their limited receptive field makes it difficult to capture long-range dependencies. As a result, purely CNN-based fusion can miss global context or higher-level semantic cues in the scene, leading to suboptimal fused outputs when important complementary information is distributed across the image.

Transformer-based fusion methods have been proposed to address the locality limitation of CNNs. By leveraging self-attention, Transformers can model long-range interactions and global context during fusion [4, 5, 6]. For example, the Image Fusion Transformer (IFT) [4] and SwinFusion [5] architectures combine CNN encoders with Transformer blocks to capture cross-modal correlations over large spatial areas, yielding more coherent fused results. While these approaches better preserve global structure and scene-level information, they still rely exclusively on the visual content of the source images and cannot leverage any external semantic knowledge beyond the pixel-level cues.

These observations have prompted a new research direction: guiding image fusion with external semantic information from vision-language models. Instead of relying solely on learned visual features, the fusion process can be informed by textual descriptions that provide high-level context about the scene. Zhao *et al.* [7] pioneered this idea with the FILM framework, which uses a large language model (ChatGPT) to generate detailed captions for the input images and then incorporates those textual features into an IR-VIS fusion network via cross-attention. The resulting fused images contain richer semantic content than previous methods. Subsequent work has also explored vision-language guided fusion by leveraging pre-trained models like CLIP to inject semantic cues into the fusion pipeline [3]. However, these early multimodal fusion approaches often introduce significant complexity and overhead (e.g. requiring lengthy text generation for each image pair or adding heavy modules for cross-modal feature alignment), which can limit their practicality.

In this work, we propose a new vision-language guided fusion model called IVTFuse, which synergistically combines multi-modal visual features with semantic textual information for infrared-visible image fusion. IVTFuse is a tri-modal network consisting of coordinated infrared, visible, and text branches. To achieve powerful yet efficient fusion, we introduce two lightweight attention modules in the IR and VIS encoders: *Frequency Strip Attention* (FSA) and *Hybrid Pooling Attention* (HPA), that adaptively emphasize informative frequency components and salient spatial regions in each modality. Meanwhile, a text encoder (we use a pre-trained BLIP model [8]) processes a descriptive caption of the scene, and the resulting text embedding is injected at multiple stages of the fusion via cross-attention. The overall architecture (illustrated in Figure 1) follows a hierarchical design: IR and VIS features are first refined by FSA and HPA, then iteratively fused with guidance from the text features across several transformer-based fusion blocks, and finally reconstructed into a single output image via a compact Restormer-based decoder [9]. Experimental results show that IVTFuse achieves state-of-the-art fusion performance.

In summary, our contributions are as follows:

- We propose **IVTFuse**, a novel *vision-language-guided* infrared-visible image fusion architecture. Unlike previous fusion methods, our tri-modal design fuses IR, VIS, and text features within a unified network, allowing high-level semantic information to directly guide the fusion process and leading to more informative fused images.

- We introduce two new lightweight attention modules, **Frequency Strip Attention (FSA)** and **Hybrid Pooling Attention (HPA)**, to enhance modality-specific feature extraction. FSA and HPA focus on complementary aspects (frequency-domain structure and spatial saliency, respectively), enabling our model to preserve fine details and important structures from each source image with minimal overhead.

- **State-of-the-art performance:** Extensive experiments on three public IR-VIS fusion benchmarks demonstrate that IVTFuse outperforms a wide range of state-of-the-art methods. In particular, our approach achieves top results across multiple quantitative metrics and produces fused images with superior detail and semantic fidelity compared to 10 recent fusion models.

## 2 Related Work

**Image Fusion Networks.** Early CNN-based methods [1, 2, 3, 10] focus on low-level feature extraction and fusion, but struggle with long-range context. Transformer-based architectures [4, 11, 12] enhance global reasoning using attention mechanisms, achieving improved performance. More recently, language-guided methods such as FILM [7] and MGFusion [3] leverage vision-language models (VLMs) to introduce semantic guidance. HSFusion [13] adopts semantic and geometric supervision via domain transformation.

**Comparison to Our Work.** Our method builds on this trend by integrating semantic cues from VLMs through a lightweight, end-to-end conditioning mechanism. Unlike prior work that relies on large generative models or separate stages, we directly inject text-guided signals into the fusion network with minimal overhead, achieving improved semantic fidelity and visual quality.

A full review of related CNN, Transformer, and language-guided fusion methods is provided in Appendix B.

## 3 Method

### 3.1 Problem Formulation

We formalize infrared-visible image fusion with language guidance as a mapping $f_\theta$ that takes an infrared image $I_{\text{IR}}$, a visible-light image $I_{\text{VIS}}$, and an associated text description $S$, and produces a single fused image $I_F$ preserving critical thermal and visual information. Formally, the fusion process can be written as:

$$I_F \ = \ f_\theta\Big(I_{\text{IR}}, \ I_{\text{VIS}}, \ E_{\text{txt}}(S)\Big), \tag{1}$$

where $E_{\text{txt}}(S)$ denotes a text encoder that produces embeddings from the input description. In our design, $E_{\text{txt}}$ is implemented using BLIP's pre-trained language model [8], which generates a sequence of textual feature tokens capturing high-level scene semantics. The goal of $f_\theta$ is to integrate multimodal information such that $I_F$ contains the complementary features of $I_{\text{IR}}$ and $I_{\text{VIS}}$ while being guided by the semantic context provided by $S$.

To do this, we propose a tri-modal fusion network, referred to here as IVTFuse, that consists of three coordinated branches (Figure 1): an IR image branch, a VIS image branch, and a text branch. The IR and VIS branches extract modality-specific features using lightweight attention modules, and the text branch injects semantic guidance via a cross-attention mechanism. The overall architecture follows a hierarchical multi-stage design. First, IR and VIS features are processed in parallel through a sequence of fusion blocks that gradually incorporate textual cues. These fusion blocks operate at a fixed spatial resolution and do not downsample, preserving full image detail. After several stages, the modality-specific feature streams are merged and passed through a series of transformer-based fusion layers to reconstruct the final fused image. This hierarchical strategy enables progressive integration: early stages align and enhance features with textual guidance, while later stages perform global feature fusion and refinement.

Concretely, IVTFuse stacks $N$ fusion blocks (we use $N = 3$ in our implementation) followed by a small fusion "decoder." Each fusion block $b_k$ processes the IR and VIS feature maps $\big(F_{\text{IR}}^{(k-1)}, \ F_{\text{VIS}}^{(k-1)}\big)$ (with $F^{(0)}$ defined as the original inputs) along with the text embeddings $T$, producing updated features $\big(F_{\text{IR}}^{(k)}, \ F_{\text{VIS}}^{(k)}\big)$. After the final block $b_N$, the two feature maps are concatenated and fed into a Restormer-based fusion decoder [9] that produces the fused image $I_F$.

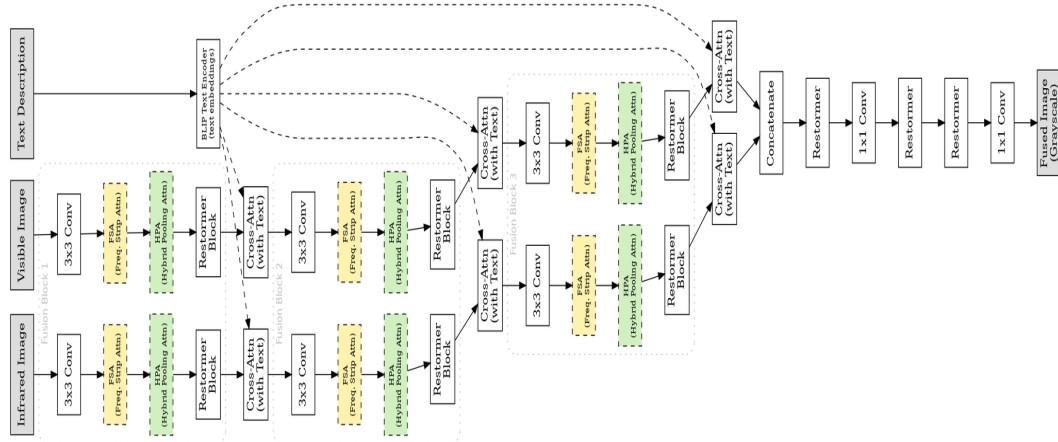

Figure 1: Flowchart of the IVTFuse architecture for infrared-visible image fusion with text guidance. The IR and VIS branches include the proposed FSA and HPA modules (highlighted) before each Restormer block. Text features (from a BLIP text encoder) guide the fusion via cross-attention in each fusion block. The decoder (right) uses Restormer layers to generate the final fused grayscale image.

## 3.2  Infrared and Visible Feature Extraction with FSA and HPA

**IR and VIS Branch Encoders:** The IR and visible images are first passed through separate encoders that share a common design. Each begins with a $3 \times 3$ convolution and PReLU activation to project the single-channel image into an initial feature map $F_{\text{IR}}$ and $F_{\text{VIS}}$ of dimension $D$ (we set $D = 32$). These feature maps then undergo two lightweight attention modules: Frequency Strip Attention (FSA) and Hybrid Pooling Attention (HPA), which adaptively refine the features by focusing on informative frequencies and spatial regions, respectively.

**Frequency Strip Attention (FSA):** This module decomposes the feature map into directional frequency components and suppresses noise while amplifying salient structure. Drawing on the dual-domain strip attention idea [14], FSA separates each feature map into low-frequency and high-frequency content along two orthogonal orientations (horizontal and vertical). Concretely, for each channel, we apply stripe-wise average pooling across rows and columns to estimate the low-frequency response in horizontal ($h_{low}$) and vertical ($v_{low}$) directions. Subtracting these from the original yields the corresponding high-frequency responses ($h_{high}$ and $v_{high}$). FSA then recombines these components with learnable weights: for example, the horizontal output is $h_{\text{out}} = w_h^{(L)} \cdot h_{low} + \left( w_h^{(H)} + 1 \right) \cdot h_{high}$, where $w_h^{(L)}$ and $w_h^{(H)}$ are learnable scalars per channel (the $+1$ ensures an initial identity mapping for high-frequency). The vertical path is computed analogously on $h_{\text{out}}$, yielding $v_{\text{out}} = w_v^{(L)} \cdot v_{low} + \left( w_v^{(H)} + 1 \right) \cdot v_{high}$. Finally, the refined feature is blended with the original input via a residual gating: $F' = \beta \cdot F + \gamma \cdot v_{\text{out}}$, with $\beta, \gamma$ learned per-channel parameters. This frequency-selective attention emphasizes important structures (e.g. edges, thermal blobs) by modulating low vs. high frequency content, while preserving the original feature as needed.

**Hybrid Pooling Attention (HPA):** The FSA-refined features next pass through HPA, which focuses on spatial information by combining global pooling strategies. HPA computes attention jointly across channel groups and spatial axes to capture a wide context with low overhead. Specifically, the feature channels are divided into $g$ groups (we use a small factor, e.g. $g = 4$, to keep the module lightweight). For each group, HPA gathers two sets of context maps: one via average pooling and one via max pooling. In the average-pooling branch, HPA pools the feature map along the horizontal direction (producing a column vector for each spatial row) and along the vertical direction (producing a row vector for each column). These capture global spatial summaries of the group's features. The pooled outputs are concatenated and passed through a $1 \times 1$ convolution, then split back into two maps corresponding to horizontal and vertical contexts. After applying sigmoid activation, these context maps act on the original features (via element-wise multiplication) to emphasize informative regions along each axis. In parallel, a complementary branch applies a $3 \times 3$ convolution (capturing local spatial context) and an analogous max-pooling pathway to obtain a second set of attention weights.

The outputs of the average and max branches are combined to yield the final attention mask for each group. This mask is applied to the group's features, and all groups are recombined. The net effect is that HPA highlights important spatial locations and structures by leveraging both average (smooth global cues) and max pooling (salient extreme cues), while also maintaining efficiency through channel grouping. After HPA, each branch has an attentive feature map $F''_{\text{IR}}$ and $F''_{\text{VIS}}$ that has been refined in both frequency content and spatial saliency.

**Local-Global Feature Transform (Restormer Block):** To further enrich each modality's features, we feed $F''_{\text{IR}}$ and $F''_{\text{VIS}}$ through a Restormer transformer block [9]. Restormer is a lightweight transformer architecture specialized for image restoration tasks, which uses multi-head self-attention and convolutional feed-forward layers to capture long-range dependencies efficiently. In our context, the Restormer block allows each modality's feature map to encode both local details and non-local context (e.g. broader thermal or color patterns) before fusion. The block preserves the feature map size and channel dimension $D$. Let $F_{\text{IR}}$ and $F_{\text{VIS}}$ denote the outputs of the Restormer encoder for the IR and VIS streams, respectively.

### 3.3 Text Embedding and Cross-Attention Fusion

**Text Feature Encoding:** The third branch of our network processes the input text description $S$ to extract embeddings that can guide the fusion. We first obtain a sequence of text tokens using BLIP's encoder [8]. This yields a sequence $T = \{t_1, \ldots, t_{L_T}\}$ of dimension $d_{\text{in}} = 768$ per token. We then project these embeddings to the fusion hidden dimension $d = 256$ using a $1 \times 1$ convolution (implemented as a 1-D linear layer) applied along the token dimension. This gives transformed text features $T' = \{t'_1, \ldots, t'_{L_T}\}$ with $t'_i \in \mathbb{R}^d$. The projection ensures the text features are compatible in size with the image feature representations. Intuitively, $T'$ encodes the semantic content of the scene in a form that our cross-attention can use to modulate visual features.

**Cross-Attention Mechanism:** We integrate the text and image features using a dual cross-attention mechanism applied to $T'$ in conjunction with the IR and VIS feature tokens. Let $A$ and $B$ denote sequences of tokens obtained by flattening the spatial feature maps $F_{\text{IR}}$ and $F_{\text{VIS}}$ (each of length $H' \times W'$ and dimension $d$). Two parallel multi-head cross-attention modules are then used: one takes the text tokens $T'$ as queries and IR tokens $A$ as keys/values, and the other uses $T'$ with VIS tokens $B$ in the same way. This operation produces attention output sequences $C_{\text{IR}} = \text{CrossAttn}(Q = T', K = A, V = A)$ and $C_{\text{VIS}} = \text{CrossAttn}(Q = T', K = B, V = B)$, each of shape $L_T \times d$. Intuitively, $\text{CrossAttn}(T', A, A)$ allows each text token to attend to all IR image tokens and pull out the visual features relevant to that word or phrase [15]. For example, a text token encoding "hot object" will assign higher attention to IR token positions corresponding to bright (hot) regions. The result $C_{\text{IR}}$ thus encodes how text elements align with IR features, and similarly $C_{\text{VIS}}$ aligns text with VIS features.

**Text-Guided Channel Reweighting:** Rather than directly merging $C_{\text{IR}}$ and $C_{\text{VIS}}$ with the image features, we use them to derive attention weights that emphasize important feature channels conditioned on the text. We average the cross-attention output across all text tokens to obtain a single vector per modality:

$$w_{\text{IR}} = \text{Norm}\Big(\frac{1}{L_T}\sum_{i=1}^{L_T} C_{\text{IR}}[i]\Big), \qquad w_{\text{VIS}} = \text{Norm}\Big(\frac{1}{L_T}\sum_{i=1}^{L_T} C_{\text{VIS}}[i]\Big), \tag{2}$$

where $C[i]$ denotes the $i$-th token's $d$-dimensional output, and $\text{Norm}(\cdot)$ is an $L_1$ normalization that scales the vector to have unit sum (effectively a softmax over the $d$ channels). Thus $w_{\text{IR}}, w_{\text{VIS}} \in \mathbb{R}^d$ act as modulation weight vectors for the IR and VIS features, respectively. Each component $w_{\text{IR}}[m]$ reflects the importance of the $m$-th feature channel in the IR tokens as signaled by the text (and similarly for $w_{\text{VIS}}[m]$ for the VIS tokens). We then re-weight the image token sequences by these vectors: every IR token $a_j$ has its $m$-th component scaled by $w_{\text{IR}}[m]$, and likewise every VIS token $b_j$ is scaled by $w_{\text{VIS}}[m]$. In matrix form, this is $\tilde{A} = \text{diag}(w_{\text{IR}}) A$ for the IR tokens (and analogously $\tilde{B} = \text{diag}(w_{\text{VIS}}) B$ for VIS). The re-weighted sequences $\tilde{A} = \{\tilde{a}_j\}$ and $\tilde{B} = \{\tilde{b}_j\}$ are then reshaped back to the 2D spatial domain (inverting the earlier flattening). This yields text-modulated feature maps $\tilde{F}_{\text{IR}}$ and $\tilde{F}_{\text{VIS}}$ of the same spatial size as $F_{\text{IR}}$ and $F_{\text{VIS}}$, but with a reduced channel count $d_{\text{proj}}$ (we use $d_{\text{proj}} = 32$ channels for the modulated features). Finally, we project these maps back to the original feature dimension $D$ using a $1 \times 1$ convolution (with PReLU) to obtain $P_{\text{IR}}$ and $P_{\text{VIS}}$, the text-guided feature maps for each modality.

**Fusion via Residual Feature Injection:** The last step within the fusion block is to inject the text-guided features into the original IR/VIS feature streams. We adopt a residual fusion approach: the projected text feature map is first added to the original feature ($P_{\text{IR}} + F_{\text{IR}}$) to enrich it with semantic cues, and this sum is concatenated with the original feature along the channel dimension. The concatenated tensor (of size $2D$ channels) is then compressed back to $D$ channels via a $1 \times 1$ convolution followed by PReLU activation. Formally, for the IR branch we compute: $F_{\text{IR}}^{(\text{out})} = \sigma\left(W_{\text{IR}}([\,F_{\text{IR}}\,;\,(F_{\text{IR}} + P_{\text{IR}})\,])\right)$, where $[\cdot\,;\cdot]$ denotes channel-wise concatenation, $W_{\text{IR}}$ are the learnable weights of the $1 \times 1$ conv, and $\sigma$ is a PReLU activation. (An identical operation is performed for the VIS branch, using $W_{\text{VIS}}$.) The result is an updated IR feature map $F_{\text{IR}}^{(\text{out})}$ and VIS feature map $F_{\text{VIS}}^{(\text{out})}$, each of size $(D,\ H',\ W')$, that now encode the original visual content augmented with text-conditioned features. This completes one cross-attentional fusion block. By design, the block preserves the spatial resolution of features and only adds a small number of channels (temporarily, during concatenation) which are immediately projected back to $D$, keeping the computation efficient.

### 3.4 Hierarchical Fusion Blocks and Final Image Reconstruction

We repeat the above fusion block process in a hierarchical series to progressively strengthen the multi-modal feature integration. The output of each block $\left(F_{\text{IR}}^{(k)},\ F_{\text{VIS}}^{(k)}\right)$ is fed as input into the next block (with the same text token sequence $T'$ reused at every stage). Stacking three such blocks ($N = 3$) proved sufficient in our experiments to fully merge the modalities. Each subsequent block operates on features that are already enriched by textual guidance and cross-modal context from previous stages, allowing deeper layers to refine more abstract representations (e.g. object-level details) under textual guidance. This design is inspired by multi-stage feature fusion networks in vision, and it ensures that information from $I_{\text{IR}}$, $I_{\text{VIS}}$, and $S$ is blended at multiple levels of abstraction.

After the final fusion block, we obtain final-stage feature maps $F_{\text{IR}}^{(N)}$ and $F_{\text{VIS}}^{(N)}$ from the two branches. At this point, each contains complementary information from both modalities (due to the cross-attention interactions) but they still reside in separate feature spaces. The fusion decoder then performs the ultimate merging and reconstruction. First, we concatenate the IR and VIS feature maps along the channel dimension, yielding a joint representation $F_{\text{cat}} \in \mathbb{R}^{2D \times H' \times W'}$. This tensor is fed through a series of Restormer blocks to perform global fusion of the IR and VIS information now combined in $F_{\text{cat}}$. We use a small stack of transformer blocks (two in our implementation) interleaved with $1 \times 1$ convolutions for channel mixing. In particular, one Restormer block operates on $F_{\text{cat}}$ (modeling long-range interactions between IR and VIS features across the entire image) [9], then a $1 \times 1$ convolution reduces the channel dimension from $2D$ back down to $D$. This is followed by another one or two Restormer blocks at $D$ channels to refine the fused feature map. Finally, a $1 \times 1$ convolution projection generates the output image $I_F$ (single channel), and a sigmoid activation is applied to scale intensities to the $[0, 1]$ range. Notably, the use of Restormer-based blocks in the decoder enables hierarchical local-global adjustments to the fused representation [9], which helps in reconstructing fine details (e.g. edges, textures) while preserving global consistency (overall contrast and semantics) in the fused image.

## 4 Experiments

**Loss Function.** We adopt a loss function widely used [16, 7]. It comprises an intensity reconstruction term and a gradient preservation term: $\mathcal{L}_{total} = \mathcal{L}_{int} + \mathcal{L}_{grad}$, where $\mathcal{L}_{int} = \frac{1}{HW} \|I_F - \max(I_{\text{IR}}, I_{\text{VIS}})\|_1$ and $\mathcal{L}_{grad} = \frac{1}{HW} \| |\nabla I_F| - \max(|\nabla I_{\text{IR}}|, |\nabla I_{\text{VIS}}|)\|_1$. Here $I_{\text{IR}}$ and $I_{\text{VIS}}$ are the infrared and visible source images, $I_F$ is the fused output, and $\nabla$ denotes the Sobel gradient operator. This loss encourages the fused image to preserve the salient intensity features and gradient details from both modalities.

**Training Details.** We implement our network in PyTorch [17] and train it from scratch using the Adam optimizer [18]. The initial learning rate is set to $1 \times 10^{-4}$, and we train for 300 epochs with a batch size of 16. The learning rate is decayed by a factor of 0.5 every 50 epochs. All experiments are conducted on an NVIDIA RTX 4090 GPU with 24 GB of memory.

**Caption Acquisition and Preprocessing.** We follow the standard setup used in [7] and adopt its text annotation pipeline without modification. Specifically, for each infrared-visible image pair, a single

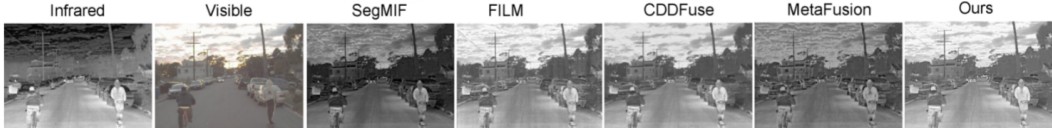

Figure 2: Visual comparison of fusion results from representative models on the infrared–visible image fusion task.

Table 1: Quantitative results of on the VLF benchmark dataset of the IVF task. The red and blue markers represent the best and second-best values, respectively.

| | MSRS Infrared-Visible Fusion Dataset | | | | | | | M³FD Infrared-Visible Fusion Dataset | | | | | | | RoadScene Infrared-Visible Fusion Dataset | | | | | |
|---|---|---|---|---|---|---|---|---|---|---|---|---|---|---|---|---|---|---|---|---|
| | EN | SD | SF | AG | VIF | Qabf | | EN | SD | SF | AG | VIF | Qabf | | EN | SD | SF | AG | VIF | Qabf |
| SDN | 5.25 | 17.35 | 8.67 | 2.67 | 0.50 | 0.38 | SDN | 6.79 | 34.63 | 14.86 | 5.16 | 0.56 | 0.54 | SDN | 7.34 | 44.74 | 14.99 | 5.94 | 0.62 | 0.55 |
| TarD | 5.28 | 25.22 | 5.98 | 1.83 | 0.42 | 0.18 | TarD | 6.79 | 40.75 | 8.18 | 2.92 | 0.53 | 0.30 | TarD | 7.25 | 47.57 | 11.46 | 4.23 | 0.56 | 0.43 |
| DeF | 6.46 | 37.63 | 8.60 | 2.80 | 0.77 | 0.54 | DeF | 6.84 | 35.09 | 9.65 | 3.37 | 0.59 | 0.42 | DeF | 7.39 | 47.60 | 11.26 | 4.47 | 0.63 | 0.50 |
| Meta | 5.65 | 24.97 | 9.99 | 3.40 | 0.63 | 0.48 | Meta | 6.68 | 29.62 | 16.22 | 5.68 | 0.68 | 0.57 | Meta | 6.87 | 31.95 | 14.40 | 5.55 | 0.55 | 0.46 |
| CDDF | 6.70 | 43.39 | 11.56 | 3.74 | 1.05 | 0.69 | CDDF | 7.08 | 41.29 | 16.49 | 5.42 | 0.78 | 0.63 | CDDF | 7.41 | 54.59 | 17.04 | 6.07 | 0.63 | 0.51 |
| LRR | 6.19 | 31.78 | 8.46 | 2.63 | 0.54 | 0.46 | LRR | 6.60 | 30.19 | 11.69 | 3.95 | 0.57 | 0.51 | LRR | 7.09 | 38.77 | 11.50 | 4.36 | 0.43 | 0.33 |
| MURF | 5.04 | 20.63 | 10.49 | 3.38 | 0.44 | 0.36 | MURF | 6.52 | 27.90 | 11.43 | 4.51 | 0.39 | 0.30 | MURF | 6.91 | 33.46 | 13.74 | 5.31 | 0.53 | 0.47 |
| DDFM | 6.19 | 29.26 | 7.44 | 2.51 | 0.73 | 0.48 | DDFM | 6.72 | 31.15 | 9.84 | 3.42 | 0.63 | 0.47 | DDFM | 7.27 | 42.94 | 10.89 | 4.20 | 0.63 | 0.50 |
| SegM | 5.95 | 37.28 | 11.10 | 3.47 | 0.88 | 0.63 | SegM | 6.89 | 35.64 | 16.11 | 5.52 | 0.78 | 0.65 | SegM | 7.29 | 47.10 | 15.07 | 5.78 | 0.65 | 0.56 |
| FILM | 6.72 | 43.17 | 11.70 | 3.84 | 1.06 | 0.73 | FILM | 7.09 | 41.53 | 16.77 | 5.55 | 0.83 | 0.67 | FILM | 7.43 | 49.25 | 17.34 | 6.60 | 0.69 | 0.62 |
| Ours | 6.73 | 43.19 | 11.72 | 3.85 | 1.09 | 0.75 | Ours | 7.10 | 42.08 | 16.69 | 5.41 | 0.91 | 0.69 | Ours | 7.46 | 50.99 | 17.35 | 6.77 | 0.69 | 0.63 |

paragraph-level textual description is generated using ChatGPT [19]. The resulting description is then encoded by the pretrained BLIP model to produce language embeddings. These BLIP features are used directly as input during both training and inference, without further processing, to ensure reproducibility and compatibility with prior work.

**Evaluation Metrics.** To quantitatively evaluate fusion performance, we use the standard metrics from the infrared-visible image fusion literature [20]. In particular, we report entropy (EN), standard deviation (SD), spatial frequency (SF), average gradient (AG), visual information fidelity (VIF), and the feature mutual information-based metric $Q_{AB/F}$. Higher values of these metrics indicate better fusion quality [20].

**Datasets.** We conduct our experiments on he VLF benchmark dataset [7], which comprises three widely-used IVF datasets: MSRS [21], M³FD [22], and RoadScene [23]. Following the protocol of FILM [7], we use the MSRS dataset for training (1083 infrared-visible image pairs) and validation (361 pairs). To evaluate generalization, the trained model is directly tested on the full test sets of M³FD and RoadScene without any fine-tuning. We compare our method against a range of state-of-the-art fusion approaches, these include SDNet [24], TarDAL [22], DeFusion [25], MetaFusion [26], CDDFuse [16], LRRNet [27], MURF [28], DDFM [29], and SegMIF [30], as well as the recent vision-language guided method FILM [7]. All methods are evaluated on the same test images using the above metrics for a fair comparison.

**Results.** Figure 2 presents the fusion results of several representative models. Compared with existing methods, our IVTFuse model more effectively integrates thermal cues from infrared imagery into the visible domain, while preserving sharp texture detail and semantic structure. Table 1 summarizes quantitative performance across the VLF benchmark for the infrared-visible fusion (IVF) task. Across all three datasets, IVTFuse consistently achieves the best or second-best performance in nearly all metrics. On the MSRS dataset, our model ranks first in **EN**, **SF**, **AG**, **VIF**, and **Qabf**, outperforming all baselines. Compared to FILM and CDDFuse, it improves both low-level contrast and high-frequency details while preserving semantic content. On M³FD, IVTFuse again leads in **EN**, **SD**, **VIF**, and **Qabf**, with competitive AG and SF, closely matching the best performers. This indicates the model's strong generalization to varied thermal-visible scenes. On the challenging RoadScene dataset, IVTFuse achieves top results in **EN**, **SF**, **AG**, **VIF**, and **Qabf**, further validating its effectiveness. Notably, our model surpasses FILM in AG and Qabf, and matches it in VIF, despite being a more modular architecture. These results demonstrate that IVTFuse's integration of frequency- and spatial-attentive modules, combined with multimodal guidance from text, enables high-fidelity fusion that preserves both global scene semantics and local detail. The consistent gains across metrics confirm its robustness and superiority over previous state-of-the-art methods.

**Ablation Studies** We conduct ablation experiments on the RoadScene dataset to quantify the contributions of the FSA and HPA modules, as well as the importance of accurate text guidance. As shown in Table 2, removing both modules (Exp. I) yields the lowest performance on all metrics, confirming

Table 2: Ablation experiment results on the RoadScene dataset (FSA/HPA modules and text guidance) with 3 random seeds. The red markers denote best values.

| Descriptions | Configurations | | Metrics | | | | | |
|---|---|---|---|---|---|---|---|---|
| | Frequency Strip Attention | Hybrid Pooling Attention | EN | SD | SF | AG | VIF | Qabf |
| Exp. I: w/o FSA and HPA | | | $7.15 \pm 0.04$ | $45.44 \pm 0.35$ | $15.30 \pm 0.29$ | $5.30 \pm 0.03$ | $0.49 \pm 0.02$ | $0.46 \pm 0.02$ |
| Exp. II: w/o FSA | | ✓ | $7.43 \pm 0.06$ | $48.57 \pm 0.42$ | $16.46 \pm 0.24$ | $5.93 \pm 0.05$ | $0.50 \pm 0.01$ | $0.53 \pm 0.03$ |
| Exp. III: w/o HPA | ✓ | | $7.42 \pm 0.03$ | $47.95 \pm 0.40$ | $16.37 \pm 0.31$ | $5.97 \pm 0.04$ | $0.58 \pm 0.03$ | $0.54 \pm 0.02$ |
| Exp. IV: with random text | ✓ | ✓ | $7.17 \pm 0.05$ | $43.67 \pm 0.33$ | $12.59 \pm 0.27$ | $5.16 \pm 0.03$ | $0.48 \pm 0.01$ | $0.45 \pm 0.02$ |
| IVTFuse (Ours) | ✓ | ✓ | $7.46 \pm 0.02$ | $50.99 \pm 0.25$ | $17.35 \pm 0.22$ | $6.77 \pm 0.03$ | $0.69 \pm 0.01$ | $0.63 \pm 0.01$ |

Table 3: Efficiency comparison between IVTFuse and FILM.

| Method | Params (M) | Inference Time (ms) |
|---|---|---|
| FILM [7] | 2.07 | 72.90 |
| IVTFuse (Ours) | 2.08 | 130.14 |

that the base network alone struggles to preserve information without our attention enhancements. Adding HPA alone (Exp. II) or FSA alone (Exp. III) leads to significant improvements, demonstrating that each module independently benefits the fusion quality. In particular, HPA boosts contrast and detail metrics such as SD ($45.44 \rightarrow 48.57$) and SF ($15.30 \rightarrow 16.46$), while FSA provides a larger gain in visual fidelity (VIF $0.49 \rightarrow 0.58$) along with a higher Qabf. Finally, the full IVTFuse model with both modules achieves the best results on all metrics (e.g., EN 7.46 and AG 6.77), outperforming the ablated variants by a clear margin. This validates that FSA and HPA are complementary: FSA's frequency-based refinement and HPA's spatial attention together yield the highest overall fusion performance.

For the language modality, we additionally conduct an experiment with *random text* inputs (Exp. IV). In this setting, each image pair is fused using an unrelated caption (randomly selected from a different image) instead of the true description. As expected, providing mismatched text significantly degrades the fusion: we observe drops in all metrics compared to using the correct captions (see Table 2). The fused output often contains artifacts or misses important objects when the guidance text is irrelevant.

**Model Complexity and Runtime.** Our IVTFuse model contains approximately **2.08** million parameters and achieves an inference speed of around **130.14 ms** per image on a $288 \times 384$ input using an NVIDIA RTX 4090 GPU. In comparison, FILM [7] has a slightly smaller model size of **2.07** million parameters and runs faster at **72.90 ms** per image. As shown in Table 3, IVTFuse introduces a moderate increase in computational cost due to the inclusion of FSA, HPA, and vision-language cross-attention mechanisms. Nevertheless, it delivers significantly improved fusion quality across benchmarks, justifying this trade-off between performance and runtime efficiency.

## 5   Conclusion

In this paper, we presented **IVTFuse**, an efficient vision-language-guided framework for infrared-visible image fusion. Our model incorporates two lightweight modules: Frequency Strip Attention and Hybrid Pooling Attention, to adaptively refine features in the frequency and spatial domains, guided by high-level semantic cues from a text description. Extensive experiments on three public datasets demonstrate that IVTFuse outperforms existing fusion methods, producing fused images with superior detail preservation and semantic fidelity. These results confirm the effectiveness of combining visual and textual modalities for image fusion and suggest promising directions for future multimodal fusion research.

## 6   Reproducibility Statement

In this work, we used ChatGPT-4o to generate the entire research narrative, while human authors conducted the experiments and validated the results. We provide the full conversation with ChatGPT-4o, documenting our prompts and the model's step-by-step generation process, at `https://chatgpt.com/share/68d1b076-f168-800f-9d2c-072f88a7b2bf`.

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

## A    Limitations and Societal Implications

While IVTFuse demonstrates strong performance, there are important limitations. First, its reliance on accurate language guidance introduces potential fragility: if the input caption is noisy, irrelevant, or adversarial, the fused output may contain artifacts or omit important thermal or visual content, as evidenced in our random text ablation. Moreover, the model is trained only on well-aligned image pairs from the VLF benchmark with human-curated captions; generalization to more diverse or noisy

data remains untested. A key direction for future work is to increase robustness by incorporating text uncertainty modeling, adversarial caption augmentation, or confidence-aware fusion mechanisms.

From a societal perspective, fusion models that integrate thermal and semantic information raise ethical and privacy concerns, particularly in surveillance or security contexts. Enhanced fusion quality, especially under poor lighting, can increase the visibility of individuals or objects in ways that may bypass consent or contextual awareness. Additionally, the use of AI-generated or human-written captions introduces risks of semantic bias: captions may unintentionally reflect stereotypes or misrepresentations that propagate into fused imagery. For instance, an over-emphasis on certain object types (e.g., "weapon," "intruder") could shift attention inappropriately during fusion.

These risks may be mitigated through several potential strategies. Caption sources should be clearly documented and, where generated automatically, optionally filtered using neutrality or safety classifiers to reduce the risk of semantic bias. Interpretability tools, such as attention visualizations, could help expose how language guidance affects fusion outcomes, enabling auditing and better understanding of the model's decision process. Future benchmarks may also incorporate evaluations focused on fairness, robustness to caption perturbation, and the impact of caption variability. While our method adheres strictly to the VLF benchmark setup, any real-world deployment should consider ethical guidelines, data governance protocols, and oversight mechanisms to ensure responsible use.

## B   Related Work

**CNN-based Image Fusion Methods.** Early deep learning approaches to image fusion predominantly employed CNN architectures for feature extraction, fusion, and reconstruction. In the absence of ground-truth fused images, many models are trained in an unsupervised or self-supervised manner, using either generative strategies (e.g. GAN or diffusion models) or discriminative (regression) strategies to blend multi-modal inputs [31]. For instance, Lu *et al.* [10] introduced a GAN-based fusion model called **GAN-HA**, which employs a heterogeneous dual-discriminator design and an attention-based generator to better preserve thermal targets and fine details. This adversarial approach improves fused image fidelity, at the cost of a more complex training process. A variety of network designs have been explored to enhance information mixing. For example, autoencoder-based fusion networks with frequent feature interactions have been used to inject more information from each source into the fused result [3]. Many works introduced architectural innovations like skip connections, dense connections, and multi-scale feature extraction to preserve details across layers and scales [1, 2]. DenseFuse [1] is a representative CNN method that uses dense skip connections to better combine infrared and visible features, mitigating information loss in deep layers. Similarly, DeepFuse [2] employs a convolutional encoder-decoder to fuse differently exposed images in an unsupervised fashion. These CNN-based methods effectively merge low-level textures and contrast from source images; however, pure CNN models often struggle to capture long-range dependencies and global context, which can limit the semantic richness of the fused output.

**Transformer-based Image Fusion Methods.** The self-attention mechanisms of Transformers have recently been exploited to address limitations of convolution-based fusion models in capturing long-range dependencies. Several works integrate Transformers into image fusion networks to enhance global context aggregation. For example, [6] proposed TransMEF, a multi-exposure image fusion framework that uses a Transformer to model cross-exposure features via self-supervised multi-task learning. [4] introduced an Image Fusion Transformer (IFT) architecture combining CNNs and Transformer blocks: local features are extracted by CNNs while a Transformer attends to global context across multi-scale features, yielding improved infrared-visible fusion results [4]. To leverage hierarchical attention, [5] developed SwinFusion, which applies Swin Transformer layers for cross-domain feature learning and achieves state-of-the-art performance on various fusion tasks by capturing long-range correlations efficiently. In the medical domain, [32] proposed a Multimodal Adaptive Transformer (MATR) for medical image fusion, employing multiscale Transformer encoders to adaptively fuse MRI and CT features at different resolutions. These Transformer-based methods demonstrate that global self-attention can substantially improve fusion quality by enriching the fused representation with broader contextual information beyond the local receptive fields of CNNs. Furthermore, Li and Wu [12] propose **CrossFuse**, a hybrid CNN-Transformer network that uses a cross-attention mechanism to enhance complementary IR and VIS features. Zhang *et al.* [11] introduce **FSATFusion**, which incorporates a frequency-spatial attention Transformer module to capture discriminative features from both modalities, achieving superior fusion performance. However,

neither method utilizes external semantic inputs (e.g. language), focusing solely on image-based information.

**Vision-Language Model Guided Image Fusion Methods.** A new line of research explores incorporating high-level semantic information from vision-language models to guide image fusion. Traditional fusion networks rely solely on visual features, but vision-language models (VLMs) provide an external source of semantic knowledge through learned image-text representations. Zhao *et al.* [7] pioneered this direction with the FILM framework, which uses a large language model (ChatGPT) to generate detailed captions for the input images and then incorporates those textual features into an IR-VIS fusion network via cross-attention. The resulting fused images contain richer semantic content than previous methods. Subsequent work has also explored vision-language guided fusion by leveraging pre-trained models like CLIP to inject semantic cues into the fusion pipeline [3]. MGFusion uses CLIP's dual vision-text encoder to inject robust semantic features: the CLIP image embeddings enrich the infrared and visible feature maps, and a CLIP-guided module then adaptively selects and fuses these features, resulting in improved detail and object highlighting in challenging scenes. By leveraging powerful pre-trained VLMs like CLIP [33], such approaches infuse fusion networks with human-aligned semantic context. Early results indicate that language-guided fusion can preserve important scene content (e.g. objects and context) more effectively than purely vision-based methods, pointing to a promising research direction for multimodal image fusion.

In addition to language-based methods, researchers have also infused high-level semantics through vision tasks. **HSFusion** [13] is a task-driven IR-VIS fusion model that generates semantic and geometric representations (via a CycleGAN-based segmentation domain transformation) to guide the fusion process. This approach can enhance fused outputs for downstream recognition tasks, but it introduces a more complex, multi-stage pipeline compared to language-guided methods.

In contrast to previous methods, our proposed fusion approach integrates multimodal information in a more tightly coupled and efficient manner. Like FILM [31], we leverage a vision-language model to provide semantic guidance, but our method does so without requiring lengthy intermediate captions or separate optimization stages. Instead, we incorporate textual cues directly into the fusion network through a lightweight conditioning mechanism, allowing the model to infuse semantic knowledge at multiple stages of visual feature processing. This design avoids the heavy reliance on external generative models (e.g. large language models) and yields a more streamlined fusion pipeline. As a result, our approach is able to harness rich semantic context from language while maintaining efficiency, distinguishing itself from prior fusion works that either remain vision-only or use multimodal cues in less integrated ways. Ultimately, our method achieves a more effective fusion of images by bridging visual and textual information in a novel architecture, leading to notable improvements over existing fusion techniques.


